# APE1 distinguishes DNA substrates in exonucleolytic cleavage by induced space-filling

Tung-Chang Liu[1,2], Chun-Ting Lin[3], Kai-Cheng Chang[2], Kai-Wei Guo[2], Shuying Wang [4,5,6,7], Jhih-Wei Chu[1,2,8,9] & Yu-Yuan Hsiao [1,2,3,8,9,10 ✉]

The exonuclease activity of Apurinic/apyrimidinic endonuclease 1 (APE1) is responsible for processing matched/mismatched terminus in various DNA repair pathways and for removing nucleoside analogs associated with drug resistance. To fill in the gap of structural basis for exonucleolytic cleavage, we determine the APE1-dsDNA complex structures displaying end-binding. As an exonuclease, APE1 does not show base preference but can distinguish dsDNAs with different structural features. Integration with assaying enzyme activity and binding affinity for a variety of substrates reveals for the first time that both endonucleolytic and exonucleolytic cleavage can be understood by an induced space-filling model. Binding dsDNA induces RM (Arg176 and Met269) bridge that defines a long and narrow product pocket for exquisite machinery of substrate selection. Our study paves the way to comprehend end-processing of dsDNA in the cell and the drug resistance relating to APE1.

[1] Institute of Molecular Medicine and Bioengineering, National Chiao Tung University, Hsinchu 30068, Taiwan. [2] Department of Biological Science and Technology, National Chiao Tung University, Hsinchu 30068, Taiwan. [3] Master's and Doctoral Degree Program for Science and Technology of Accelerator Light Sources, National Chiao Tung University, Hsinchu 30068, Taiwan. [4] Department of Microbiology and Immunology, College of Medicine, National Cheng Kung University, Tainan, Taiwan. [5] Center of Infectious Disease and Signaling Research, National Cheng Kung University, Tainan, Taiwan. [6] Institute of Basic Medical Sciences, College of Medicine, National Cheng Kung University, Tainan, Taiwan. [7] Department of Biotechnology and Bioindustry Sciences, College of Bioscience and Biotechnology, National Cheng Kung University, Tainan, Taiwan. [8] Institute of Bioinformatics and Systems Biology, National Chiao Tung University, Hsinchu 30068, Taiwan. [9] Center For Intelligent Drug Systems and Smart Bio-devices (IDS2B), National Chiao Tung University, Hsinchu, Taiwan. [10] Drug Development and Value Creation Research Center, Center for Cancer Research, Kaohsiung Medical University, Kaohsiung, Taiwan. ✉email: mike0617@nctu.edu.tw

Participating in DNA repair and responding to oxidative stresses in the cell, apurinic/apyrimidinic endonuclease 1 (APE1) is a multifunctional enzyme in maintaining genome integrity. DNA repair is associated with anticancer drug resistance, counteractant of radiotherapy, tumor aggressiveness, and poor prognosis, and inhibition of APE1 is thus a promising strategy for developing antitumor drugs[1–3]. Currently, several inhibitors that repress the nuclease activity of APE1 are under evaluation in different phases of clinical trials[1–6]. APE1 exerts its function in DNA repair through the nuclease domain which possesses both the endonuclease and the 3′–5′ exonuclease activity[7,8]. In the base-excision repair (BER) pathway, APE1 acts as an apurinic/apyrimidinic (AP) site-specific endonuclease to initiate the repairing of common DNA lesions (uracil, alkylated and oxidized bases, and AP site) by incising the phosphodiester backbone at the damaged site[9–12]. The nuclease domain of APE1 also has the 3′–5′ exonuclease activity[8,13–16], which is linked to DNA mismatch repair[17], nucleotide incision repair (NIR)[18], tri-nucleotide repeat (TNR) expansion-related BER[19], DNA single-strand breaks (SSB)[6], removal of 3′-blocking groups in a nucleotide excision repair (NER)-independent pathway[20,21], and apoptosis[22,23]. However, the structural basis for the exonucleolytic actions of APE1 is currently lacking.

As an exonuclease, APE1 digests both matched and mismatched 3′-termini of a duplex DNA with a nick, gap, or recessed (5′-overhang) structures; the matched 3′-end is involved in numerous physiologically important pathways and is more resistant to the enzyme[13–16,20,24,25]. APE1 can directly interact with the error-prone DNA polymerase β[26–28] and may provide the necessary 3′–5′ exonuclease activity for proofreading and correcting mistakes during DNA synthesis[8,15,29]. Although the structural basis for APE1 excising a mismatched nucleotide (1 nt) has been addressed partially[17], whether cleaving a matched 3′-terminus adopting a different principle is unclear. A unified view is also necessary to understand the lack of base preference in APE1's exonuclease activity. Furthermore, whether APE1 can digest 3′-end with longer mismatches is unknown. To solve these puzzles, we systematically design an array of substrates for measuring the exonuclease activity and substrate binding of APE1 and also provide unique structural information for exonucleolytic complexation of matched dsDNA at the terminal.

Excision of the 3′-matched base pair by APE1 is involved in many important pathways of DNA repair, including NIR[18], TNR expansion-related BER[19], SSB[6], and apoptosis[22,23]. The 3′–5′ exonuclease activity allows APE1 to prevent the formation of lethal double-strand breaks when repairing bi-stranded clustered DNA damage by NIR[30,31], a DNA glycosylase-independent DNA repair pathway that is responsible for repairing the oxidative damage on nucleotides[31]. TNRs are present in a wide range of genes and the expansion of TNRs affects genome stability[32,33]. TNR expansions involve the 3′-matched base pair[19] and result in the formation of non-B form DNA structures, including hairpins, triplex, and stick DNAs that are associated with over 40 human neurodegenerative diseases, such as spinobulbar muscular atrophy (SBMA), Huntington disease (HD), and spinocerebellar ataxias (SCAs)[32]. APE1 has been demonstrated to prevent TNR expansions via its 3′–5′ exonuclease activity to promote the removal of hairpin structure[19]. In addition, APE1 can sense and recognize SSB sites, the most abundant form of DNA damage and performs the exonucleolytic digestion for initiating the SSB repair[6]. APE1 is also reported as an apoptotic nuclease and involves in the exonucleolytic digestion of chromosomal DNA fragments[22].

In contrast to the endonuclease activity that is highly specific to the AP site, the 3′–5′ exonuclease activity of APE1 does not display base specificity and can even remove the abnormal deoxynucleosides generated by oxidative stress, ionizing radiation, and medicine[8]. Among these, 8,5′-cyclo-2′-deoxyadenosine (cdA) and 8,5′-cyclo-2′-deoxyguanosine (cdG) are associated with genome instability and diseases. Both cdA and cdG cannot be repaired by NER, thus requiring APE1 for removal[20]. Moreover, 7,8-dihydro-8-oxodeoxyguanosine (8-oxoG) is a major mutagenic DNA lesion and has been shown to be repaired by the APE1 homolog in Saccharomyces cerevisiae, Apn1, in an alternative DNA repair pathway[21]. In addition to abnormal deoxynucleosides, the base nonspecificity of the exonuclease activity of APE1 can also remove nucleoside analogs used in anticancer or antiviral therapies[16,34]. In this regard, β-L-dioxolane-cytidine (L-OddC) is a deoxynucleoside analog with L-configuration used in anticancer therapy and is under phase III clinical trial showing promising activity for treating leukemia[16,34]. Another group of examples includes 3′-azido-3′-deoxythymidine (AZT) and 2′,3′-didehydro-2′,3′-dideoxythymidine D4T that are used as anti-retroviral agents for prevention and treatment of human immunodeficiency virus/acquired immune deficiency syndrome (HIV/AIDS)[15]. Removal of these therapeutic deoxynucleoside analogs[15,16,34] by APE1 may lead to drug resistance, highlighting the importance of understanding the 3′–5′ exonuclease activity of APE1 in anticancer and antiviral therapy.

Even though the excision of 3′-matched nucleotide by APE1 is biologically significant, the structural basis of APE1 in unwinding and digesting the base pair in various duplex DNA structures is not available. In particular, how does APE1 non-specifically cleaves abnormal bases at the 3′-end in dsDNA is mysterious. To firmly establish the distinction between the endo- and exonucleolytic cleavage of APE1, we synthesize an array of dsDNAs with bases varied in the middle or at the 3′-end for assaying the nuclease activity. Next, by resolving the structures of such substrates bound with APE1 using X-ray crystallography and structural modeling based on an empirical force field of molecular mechanics, we establish that the product pocket in the complex can accommodate various DNA bases without any specific interactions. This unprecedented structural information shows how APE1 non-specifically accommodates various bases at the 3′-end of dsDNA. To systematically analyze the structural preference of APE1's exonuclease activity, we also synthesize matched and mismatched dsDNA, recessed dsDNA, gapped dsDNA, and nicked dsDNA for assaying the nuclease activity and binding affinity with APE1. By resolving the first APE1 structures in exonucleolytic binding at dsDNA terminal, an induced space-filling model is deduced for the molecular mechanism by which APE1 distinguishes the structures of different dsDNA substrates. This principle is consistent with the available APE1 structures collected thus far and provides a unified view for the mechanism of digesting duplex DNA substrates with matched and mismatched 3′-end. Our study thus conveys the molecular principles for the in vivo functions of APE1 and for pursuing the related biomedical applications.

## Results

**Exonuclease activities of APE1.** The recombinant full-length mouse APE1 (mAPE1) and truncated mAPE1 mutants are purified into a homogeneous state with high purity (Supplementary Fig. 1a) for biochemical and structural analysis. To determine the catalytic properties and substrate preference in the endo- and exonucleolytic mode of action, we design and synthesize different dsDNAs with bases varied in the middle or at 3′-end (Supplementary Table 1). We use a 20-nucleotide (20-nt) segment of ssDNA, a 20 base-pair (20 bp) blunt-ended dsDNA, and AP site-containing dsDNA as substrates for investigating and comparing the exonuclease and endonuclease activities of APE1. The results

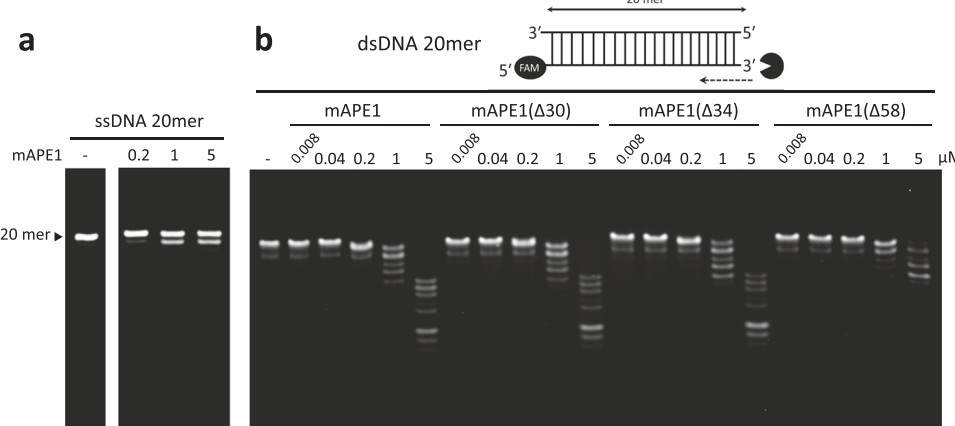

**Fig. 1 Catalytic properties of mAPE1 in exonucleolytic cleavage. a**, **b** Substrate preference of mAPE1 in digesting ssDNA and dsDNA. Higher activity in digesting dsDNA substrates is observed for mAPE1. The activities of full-length and truncated mAPE1 display similar activity levels toward dsDNA, except mAPE1(Δ58) which shows a significant reduction in dsDNA degrading activity. **a**, **b** Source data are provided as a Source Data file.

establish that the full-length mAPE1 is able to exonucleolytically digest the blunt-ended dsDNA from 3′- to 5′-end in a concentration-dependent manner (Fig. 1b). In contrast, mAPE1 displays very low activity in digesting ssDNA (Fig. 1a). To understand the preferred conditions for the endo- and exonuclease activity of mAPE1, we examine the effects of magnesium chloride (MgCl$_2$) and sodium chloride (NaCl) concentrations. The results show that MgCl$_2$ is essential for both the endo- and exonuclease activity. In particular, mAPE1 exhibits higher endo- and exonuclease activity in the presence of NaCl at the concentration of 120 mM and 30 mM, respectively (Supplementary Fig. 1b). These optimal conditions are used in the following endo- and exonuclease activity assays. We also examine endo- and exonuclease activity of various truncated mAPE1 mutants which also play roles in various important cellular events, including APE1 with truncation of 30 amino acids at the N-terminal (mAPE1Δ30), which is involved in granzyme A-mediated apoptosis[23]. The mAPE1 with deletion of 34 amino acids at the N-terminal (mAPE1Δ34) is in apoptosis[22], and the mAPE1 with truncated redox domain (mAPE1Δ58)[8] is also investigated. The results show that full-length mAPE1, mAPE1Δ30, and mAPE1Δ34 all exhibit similar levels of endo- and exonuclease activity (Fig. 1b and Supplementary Fig. 2). For mAPE1Δ58, on the other hand, lower exonuclease activity is observed, suggesting that the redox domain of mAPE1 also plays a role in affecting the exonucleolytic cleavage.

**Exonucleolytic cleavage of mAPE1 is base nonspecific**. To understand the base preference of APE1, we synthesize blunt-ended dsDNAs containing an AP site, 8-oxoG, or hypoxanthine (deoxyinosine, I base) in the middle for measuring the endonuclease activities as the baseline. For quantifying the exonuclease activities, the blunt-ended dsDNAs with the AP site or 8-oxoG locating at the 3′-end are synthesized. The results in Fig. 2 show that the endonuclease activity of APE1 is in general higher than that of exonucleolytic cleavage. The products of endonucleolytic digestion appear as the enzyme concentration is higher than 0.04 μM, while a higher than 0.2 μM of mAPE1 concentration is required for the exonucleolytic products to show up. In assaying the endonuclease activity toward the aforementioned substrates, mAPE1 only incises the AP site-containing dsDNA, indicating the very specific base preference of the endonuclease activity (Fig. 2a). Interestingly, exonucleolytic digestion after the endonucleolytic cleavage of the AP site-containing dsDNA is observed, showcasing that APE1 is able to unwind the matched DNA base

pair and remove one base at a time from the 3′-end (Fig. 2a, lanes 9–10). In assaying the exonuclease activity against the different substrates stated earlier, mAPE1 removes all dsDNA substrates with a modified base at the 3′-end, including AP site and 8-oxoG. Surprisingly, mAPE1 also removes the non-DNA substrate, biotin, further illustrating the lack of base specificity in exonucleolytic catalysis (Fig. 2b).

**Exonucleolytic cleavage of APE1 distinguishes substrate structures**. To analyze if the exonucleolytic digestion of APE1 would vary for dsDNA substrates of different structures, we synthesize blunt-ended dsDNA, 3′-end recessed dsDNA with different 5′-overhang lengths, 1 nucleotide (nt) gapped dsDNA, and nicked dsDNA that contains matched or mismatched (1 nt and 2 nt) 3′-terminus for measuring the nuclease activity. These structural dsDNA substrates mimic the DNA intermediates involved in various DNA repair pathways or apoptosis. For example, gapped dsDNA would appear in the TNR expansion-related BER[19], SSB[6], and NIR[31]; blunt-ended or recessed dsDNA would come out of DNA fragmentation during apoptosis[22]. The results are summarized in Table 1. Both blunt-ended dsDNA and dsDNA with 1-nt mismatch can be digested by mAPE1 with the former showing slightly higher resistance. On the contrary, mAPE1 exhibits no detectable activity in digesting dsDNA with 2-nt mismatch (Fig. 3a). This result is a first observation that exonucleolytically, APE1 can only cleave matched or mismatched dsDNA with the 3′-overhang shorter than two nucleotides. When targeting matched recessed dsDNAs, mAPE1 shows that the activity increases with the length of the 5′-overhang: 20-nt 5′-overhang >5–nt 5′-overhang >without 5′-overhang (Fig. 3). The activity in digesting matched recessed dsDNA with 20-nt 5′-overhang is very close to that in cleaving recessed dsDNA with 1-nt mismatch, indicating that 5′-overhang length is also a structural property affecting APE1 activity. Very low activity, though, is still observed for mAPE1 in digesting recessed dsDNA with 2-nt mismatch (Fig. 3c). The results show that the length of both 3′- and 5′-overhang on the dsDNA substrate are important structural factors for the exonuclease activity of APE1, and shorter 3′-overhang (<2 nt) and longer 5′-overhang are favored. In addition, mAPE1 shows similar catalytic power against the 1-nt-gapped dsDNA as the other substrates. Similarly, this gapped dsDNA with 1-nt mismatch is slightly more vulnerable and the 2-nt mismatch version is barely digested by mAPE1 (Fig. 3d).

In digesting the nicked dsDNA, whether the 5′-end is terminated by a phosphoryl or hydroxyl group has been shown

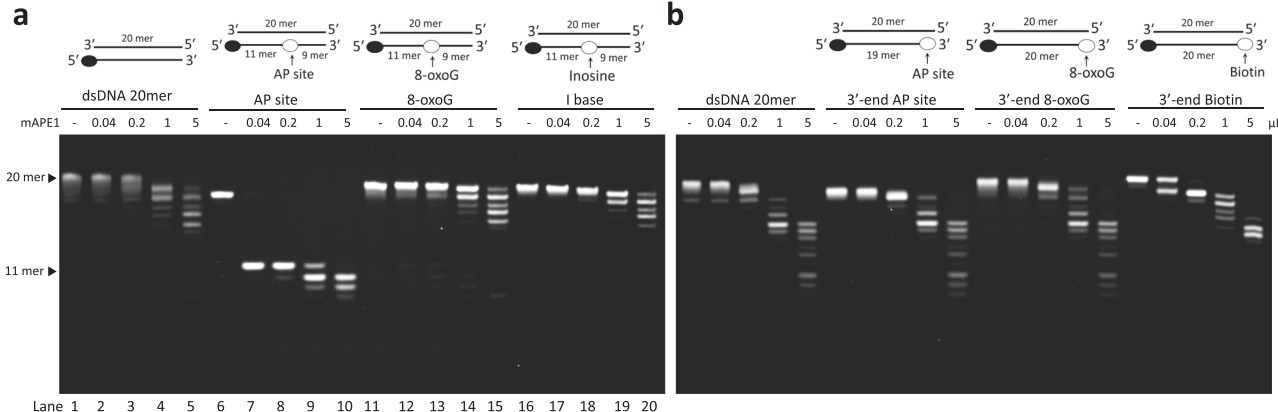

**Fig. 2 Base preference of mAPE1 in endo- and exonucleolytic cleavage. a** Incubation of mAPE1 with 5′-FAM-labeled dsDNA substrates, including perfectly paired dsDNA (dsDNA 20 mer), dsDNA with an AP site in the middle (AP site), dsDNA with a 8-oxoguanine in the middle (8-oxoG) and dsDNA with a hypoxanthine base in the middle (I base). The endonuclease activity of mAPE1 demonstrates high specificity, only AP site containing dsDNA is incised. The NaCl salt concentration for the endonuclease activity assays is 120 mM. **b** In exonuclease activity assays, mAPE1 is incubated with 5′-FAM-labeled dsDNAs containing different nucleotide modification or biotin at the 3′-end, including AP site (3′-end AP site), 8-oxoG (3′-end 8-oxoG), and biotin (3′-end biotin). All the dsDNAs with different 3′-end modifications are excised by mAPE1 as the protein concentration is higher than 0.2 μM. The NaCl salt concentration for the exonuclease activity assays is 30 mM. **a**, **b** Source data are provided as a Source Data file.

| | Blunt-ended dsDNA | Recessed dsDNA | | Gapped dsDNA | Nicked dsDNA | |
|---|---|---|---|---|---|---|
| **Table 1 The exonuclease activity of mAPE1 in processing various dsDNAs.** | | | | | | |
| | | 5-nt 5′-overhang | 20-nt 5′-overhang | | 5′-phosphoryl group | 5′-OH |
| Matched | + | + + | + + + | + + + | + | + + + |
| 1-nt mismatch | + + | + + + | + + + | + + + | + + + | + + + |
| 2-nt mismatch | − | | − | − | | |

The four activity levels for APE1 on digesting DNA substrates are represented by (−), (+), (+ +), and (+ + +). (−): cleavage pattern cannot be observed. (+), (+ +), and (+ + +): the cleavage pattern can be observed at the mAPE1 concentration of 0.75, 0.5, and 0.25 μM, respectively. Space means the condition was not tested.

to affect the exonuclease activity of APE1[14,25]. By using our exonuclease activity assay, mAPE1 shows the consistent result of higher activity against the nicked dsDNA with 5′-hydroxyl than that against the nicked dsDNA with 5′-phosphoryl (Fig. 3e). The 5′-hydroxyl nicked dsDNA is degraded at the 0.25 μM concentration of APE1, but the 5′-phosphoryl version requires a higher concentration of 0.5–0.75 μM (Fig. 3e). For 1-nt-mismatched nicked dsDNA, on the other hand, the dependence of APE1's exonucleolytic cleavage on the 5′-end drastically reduces, and mAPE1 only has slightly higher activity in digesting the 1-nt-mismatched nicked dsDNA with 5′-hydroxyl (Fig. 3e).

**Product-bound crystal structures of mAPE1 with blunt-ended dsDNA and recessed dsDNA.** To reveal the molecular mechanism by which APE1 conducts terminal processing of dsDNA and distinguishes different substrate structures, we determine two crystal structures of mAPE1 complexing dsDNA products, including mAPE1 bound to a blunt-ended dsDNA and to a recessed dsDNA (Fig. 4a, b). Both structures reveal the unprecedented atomic details of exonucleolytic cleavage for mAPE1 in a terminal-binding mode. The crystallization conditions, data collection, and refinement statistics are listed in Supplementary Tables 2 and 3.

In the crystal structure of mAPE1 blunt-ended dsDNA, the 39 residues at the N-terminal are missing due to degradation during the crystallization process. This event is evidenced by SDS-PAGE and western blot analyses (Supplementary Fig. 3a). In addition, the last nucleotide at the 3′-end of the input DNA is removed by mAPE1 during crystal formation (Fig. 4b) in the mAPE1

blunt-ended dsDNA structure as indicated by comparing the fragment size of [γ32P]ATP isotope-labeled dsDNA in the crystal with the input dsDNA sample (Supplementary Fig. 3b). Therefore, mAPE1 still exhibits the exonuclease activity under the crystallization condition (Supplementary Fig. 3c), suggesting that the necessary divalent ion may come from the *E. coli* host. During the long process of crystal formation (3 to 5 weeks), a trace amount of magnesium ion would be sufficient to render the observed removal of the last nucleotide.

In the mAPE1 blunt-ended dsDNA structure, the omit map of the double-strand region in dsDNA can be clearly visualized (Supplementary Fig. 4a) and the two 3′-ends insert into the active sites of the two mAPE1 molecules. These newly solved end-binding structures provide a unique opportunity to shed light on the very complicated mechanism of APE1 by comparing with the available structures of APE1 complexed with dsDNA that together, reveal the structural information of middle binding. Alignment of our mAPE1 blunt-ended dsDNA structure with the human APE1 (hAPE1)-mismatched dsDNA substrate complex (PDB entry: 5WN5) indicates that the last nucleotide at the 3′-end in our structure is indeed missing and shows that two crystal structures resolved here are product-bound complexes of APE1 (Fig. 4c).

Orientations of the active residues, particularly Asp69, Glu95, Tyr170, Asp209, Asn211, Asp307, and His308 in the terminal-binding mAPE1 blunt-ended dsDNA structure, are similar to those in the 5WN5 structure, in which hAPE1 binds the dsDNA substrate in the middle (Fig. 4d). After introducing the E95A and H308A mutation to the active site mAPE1, we find that both the endo- and exonuclease activity of mAPE1 reduce drastically

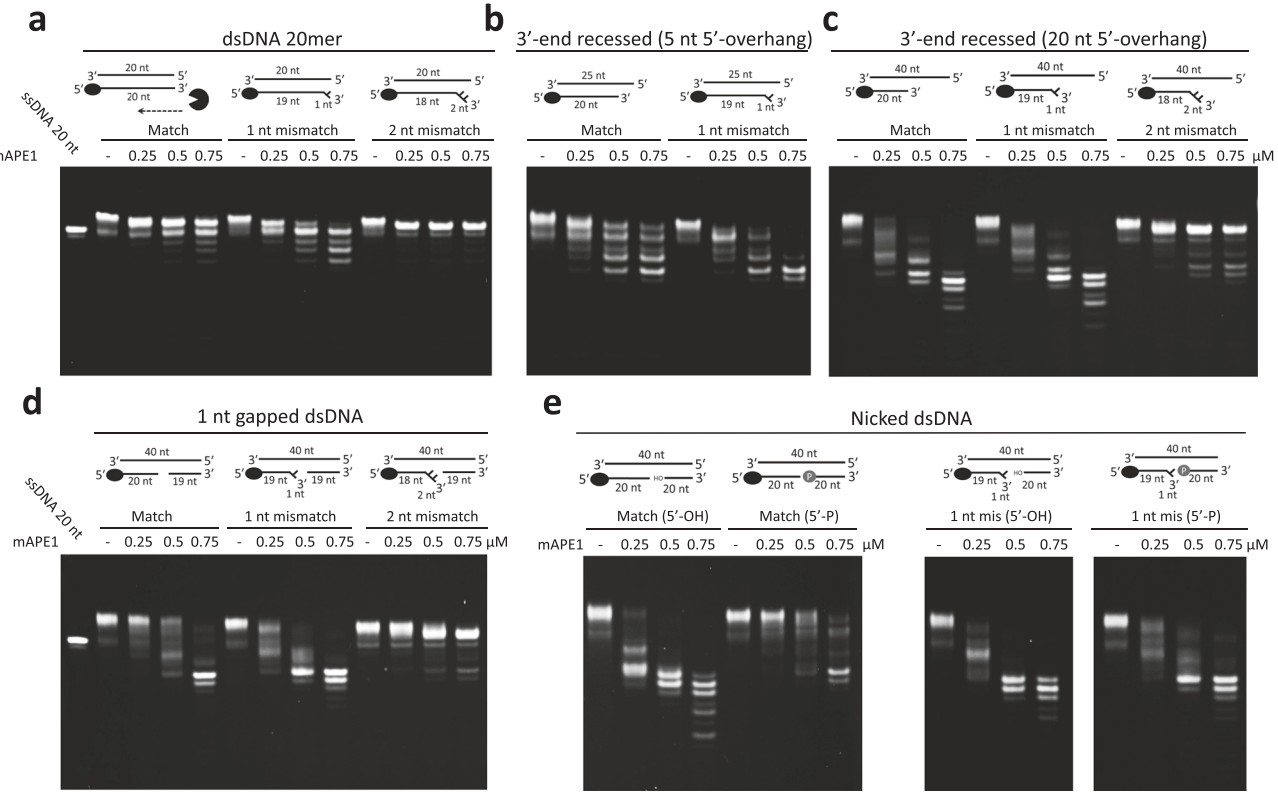

**Fig. 3 Structural preference of mAPE1 in exonucleolytic cleavage of various dsDNAs. a** The exonuclease activity of mAPE1 in digesting blunt-ended dsDNA 20 mer, dsDNA 20 mer with 1-nt mismatched base pair, and dsDNA 20 mer with 2-nt mismatched base pair. **b, c** The exonuclease activity of mAPE1 in digesting recessed dsDNA. The substrates are matched or 1-nt mismatched recessed dsDNAs with 5-nt 5′-overhang and matched or mismatched (1 nt and 2 nt in the length) recessed dsDNAs with 20-nt 5′-overhang. **d** The exonuclease activity of mAPE1 in digesting 1-nt-gapped dsDNAs. The gapped DNAs are matched, or contain 1- or 2-nt mismatched base pair at the 3′-terminus of the gapped site. **e** The exonuclease activity of mAPE1 in digesting matched or 1-nt mismatched nicked dsDNA. The nicked dsDNAs are with phosphoryl or hydroxyl groups at the 5′ margin. **a–e** Source data are provided as a Source Data file.

(Supplementary Fig. 5), indicating the importance of these residues in delivering both types of nuclease activity.

In addition, the crystal structure of mAPE1-recessed dsDNA is also determined with the substrate sequence identical to that in the mAPE1 blunt-ended dsDNA structure except for the longer 5′-overhang (Fig. 4b). Interestingly, the superposition of the two structures indicates that the two end-binding dsDNAs fit very well in the region near the active site (Fig. 4b). However, the structures of the 5′-overhang segment and over the region 4-5 base pair away from the active site of the bound products cannot be superposed (Supplementary Fig. 4b, c). This result indicates that the 5′-overhang affects the binding mode of mAPE1 with the dsDNA substrates.

**Product pocket of APE1 can accommodate normal nucleotides and various damaged DNA bases in the exonucleolytic binding mode.** The interaction maps between mAPE1 and dsDNA in mAPE1 blunt-ended dsDNA and in mAPE1-recessed dsDNA structures are shown in Supplementary Fig. 6. In both cases of terminal binding, a channel-like pocket is observed in the active site of mAPE1 to accommodate the leaving group (Fig. 5a, b). The channel is formed by Arg176 and Met269 in two separate loops, which interact in a bridge-like fashion (the RM bridge) across the active site. In the middle-binding structures of APE1 with mismatched and AP site containing dsDNA[11,17,35], the corresponding hMet270 and hArg177 residues arrange in a similar manner, but it is unclear if it could be retained in the terminal-binding

mode and applicable to process a more recalcitrant, matched dsDNA substrate exonucleolytically.

The omit map of Arg176 and Met269 in mAPE1 blunt-ended dsDNA complex and mAPE1-recessed dsDNA complex can be clearly visualized (Supplementary Fig. 7). Formation of the RM bridge is induced by protein–DNA interactions since Arg176 interacts with three dsDNA bases via hydrogen bonding as revealed in both the mAPE1 blunt-ended dsDNA and mAPE1-recessed dsDNA structures, and the RM bridge is not present in any of the apo-APE1 structures in the Protein Data Bank (PDB) (Fig. 5b). These three bases include the orphan base, the last nucleotide at the 5′-end of the non-scissile strand, and the last nucleotide at the 3′-end of the scissile strand. Similar interactions are also observed in the APE1–DNA structures deposited in PDB (middle-binding mode; Supplementary Fig. 8).

The APE1 structures in terminal binding suggest a key functional role of the RM bridge in exonuclease activity, separation of the scissile and non-scissile strands to make the orphan base away from the active site and to push the last nucleotide at the 3′-end into the product pocket for cleavage (Fig. 5a, c). The product pocket is composed of a group of residues, including Asn225, Asn228, Ala229, Phe265, Thr267, Trp279, and Leu281 (Supplementary Fig. 9a, b), and is spatially sufficient to accommodate an entire nucleotide. This product pocket can also accommodate a mismatched 3′-end of the scissile strand (Fig. 5c, the 5WN5 structure). Furthermore, the product pocket defined by RM bridge has enough space for the AP site and the downstream phosphoryl group in endonucleolytic

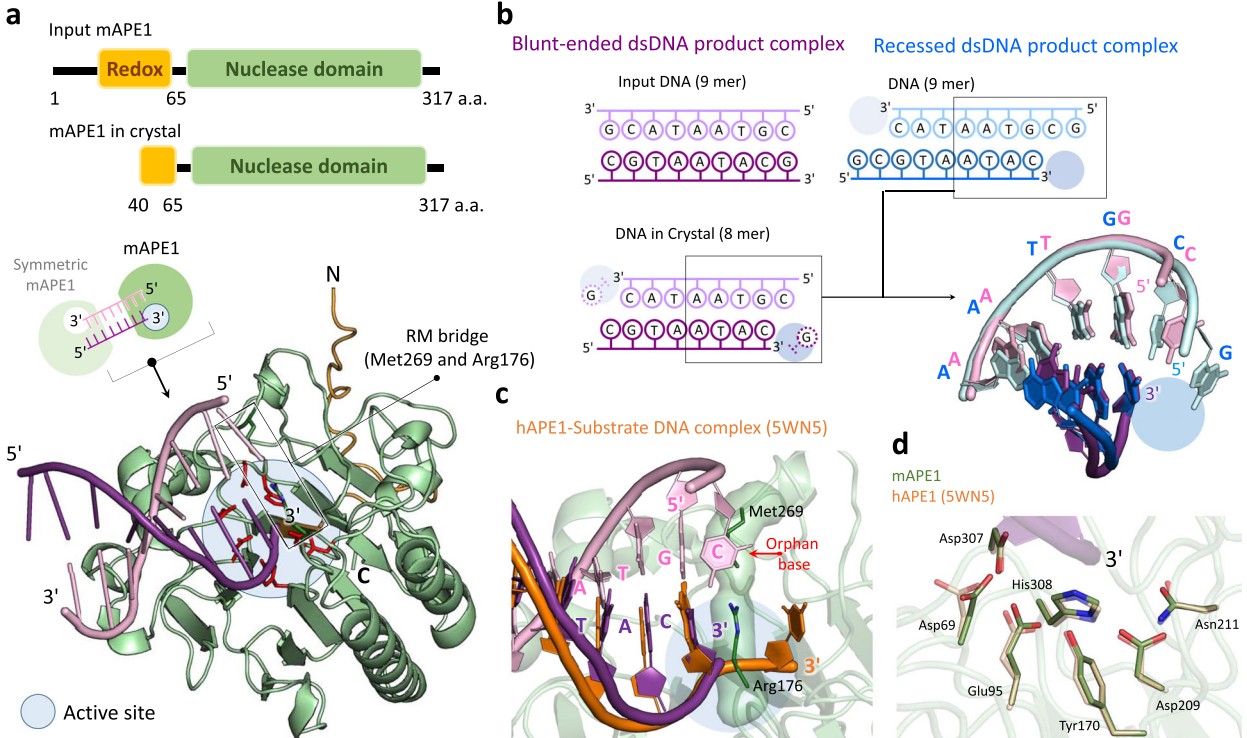

**Fig. 4 Structures of mAPE1 blunt-ended dsDNA product complex and mAPE1-recessed dsDNA product complex. a** The domain structure, crystal packing and overall structure of the mAPE1 blunt-ended dsDNA product complex. The redox domain and nuclease domain are colored in orange and green, respectively. **b** The schematic presentation of the two dsDNAs in the two product complex structures. Bottom right panel: structural alignment of the two dsDNAs in the two product complex structures. The active sites are highlighted by blue circles. **c** Structural alignment of mAPE1 blunt-ended dsDNA product complex and hAPE1-substrate DNA complex (PDB entry: 5WN5). It can be seen that the last nucleotide in the 3′-end of dsDNA in our product complex is missing. Only the scissile strand of dsDNA in the hAPE1-substrate DNA complex (colored in orange) is displayed. **d** Superposition of active sites in mAPE1 blunt-ended dsDNA product complex and hAPE1-substrate DNA complex. The active site residues in the hAPE1-substrate DNA complex are colored by light orange and displayed in a transparent mode.

cleavage (Fig. 5e, the APE1-AP site-dsDNA structure, PDB entry: 5DFI), but the presence of base would cause a steric clash, hence explaining the molecular origin for the AP-site specificity of APE1 as an endonuclease.

To illustrate that the product pocket in our mAPE1 structures of terminal dsDNA binding is spatially sufficient not only for normal bases but also for a variety of damaged nucleotides, structural models are constructed using the CHARMM empirical force field[36] for adding the 3′-terminal nucleotide back in the mAPE1 blunt-ended dsDNA structure with several forms of damaged bases, including 8-oxoG, cdA, L-O-ddC and AZT that are known substrates of APE1[15,16,20,21,34]. These structural models indeed demonstrate that the APE1 product pocket is able to accommodate all of the tested nucleotides without encountering any steric hindrance (Fig. 5d, Supplementary Fig. 9). Our structural models show that the product pocket places the last nucleotide of the substrate without hydrogen bonding or other specific interactions with APE1, hence explaining the lack of base specificity in the exonuclease activity.

**Consistent interaction zone in dsDNA for endo- and exonucleolytic cleavage by APE1.** To understand how the APE1 binding site recognizes various substrate structures, we systematically analyze the available structures in PDB for APE1–DNA complexes. The dsDNAs in these structures can be categorized into three groups, middle binding of dsDNA containing an AP site (substrate and product; PDB entry 5DFI and 5DFF), middle binding of mismatched gapped or nicked dsDNA (substrate and

product; PDB entry 5WN5 and 5WN1), and our terminal binding of blunt-ended and recessed dsDNA. Superposition of the dsDNA structures with APE1 shows a clear consistent interaction zone (Fig. 6a, b) over which the different substrate structures can fit very well with each other (colored in yellow in Fig. 6b). Such conserved hot spots for substrate recognition include the scissile strand ribose-phosphate backbone of the flank region near the cleavage site and the non-scissile strand ribose-phosphate backbone of three consecutive nucleotides locating at two bases downstream of the orphan base that have specific hydrogen bonds with APE1 (Fig. 6). As summarized in Supplementary Fig. 8, the RM bridge bounds the consistent interaction zone, and Arg176 interacts with several bases near the active site, including the orphan base and the last paired nucleotide at the scissile strand 3′-end.

Outside the consistent interaction zone, on the other hand, the structures of APE1–DNA complexes reveal divergent protein–DNA interactions. Among the three dsDNA categories mentioned earlier, our mAPE1-recessed dsDNA structure shows the unique positioning of the extended area of the 5′-overhang. For APE1, the mode of binding recessed dsDNA is thus different from that of binding dsDNA with an AP site or a mismatched base pair (Supplementary Fig. 10a).

With this structural basis, we conduct binding affinity measurements of APE1 to elucidate the effects of different substrate structures on binding. For dsDNAs with the same length in scissile and non-scissile strands, 20-bp dsDNA with 1- or 2-nt mismatch is synthesized for the electrophoretic mobility shift assay (EMSA) of binding with APE1. As a controlled

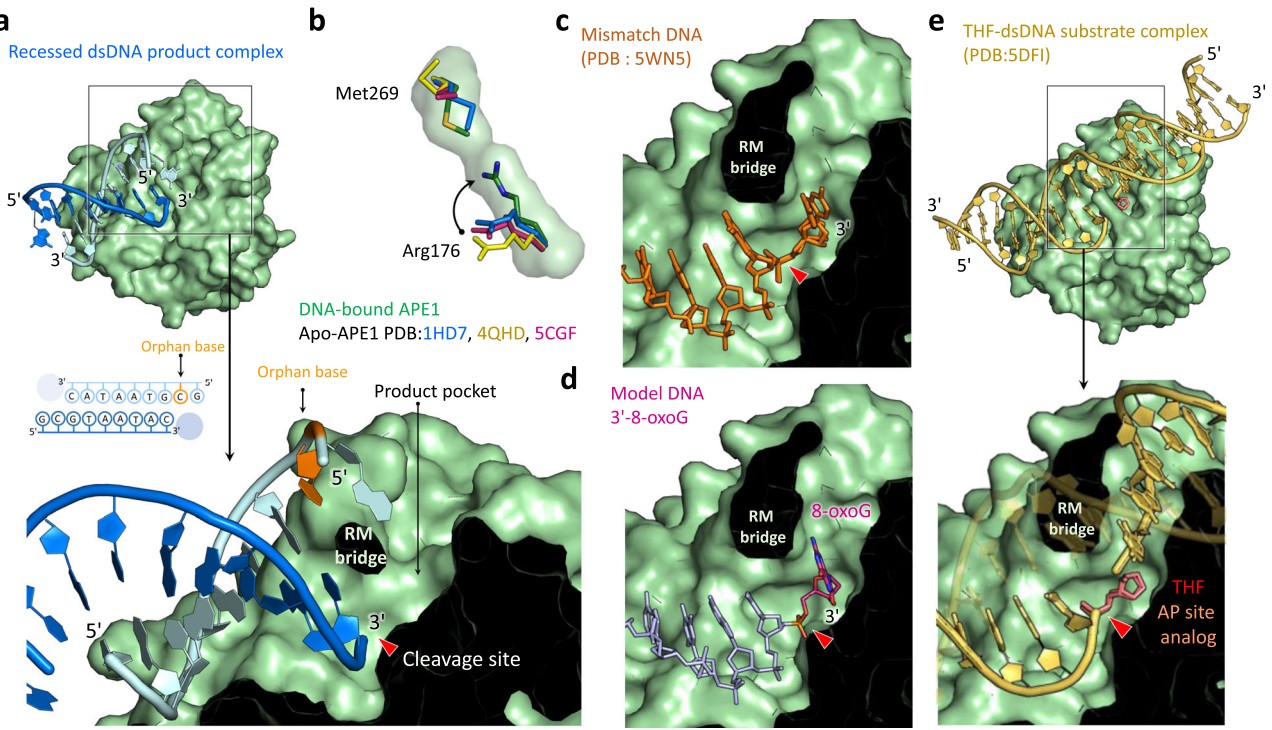

**Fig. 5 The structures of RM bridge and product pocket in the active site of APE1. a** The close-up view of the active site in mAPE1-recessed-dsDNA product complex, including the terminal of the dsDNA product, the RM bridge, and the product pocket. **b** Structural alignment of Arg176 and Met269 in apo- or DNA bound APE1 structures. The PDB entry of apo-APE1 structures are 1HD7, 4QHD, and 5CGF. **c–e** The close-up view of a normal cytosine base, an AP site analog, and a damaged 8-oxoG base in the product pocket. The cytosine coordinates are extracted from the hAPE1-mismatched dsDNA substrate complex structure (PDB entry: 5WN5). The AP site analog coordinates are extracted from the hAPE1 AP site analog-containing dsDNA complex structure (PDB entry: 5DFI). The coordinates of the damaged 8-oxoG base are generated by the molecular modeling conducted in this study.

comparison, 20-nt ssDNA is also synthesized for the EMSA measurement. Without the consistent interaction zone in ssDNA, its binding affinity with mAPE1 is clearly lower than that of dsDNA substrates, and only a few mAPE1-ssDNA complexes appear in the gel (Supplementary Fig. 10b). For the impact of having mismatched base pairs, the band of 2-nt mismatch dsDNA complexing with mAPE1 is similar to that of blunt-ended dsDNA and 1-nt mismatch dsDNA (Fig. 6c), suggesting that the common interaction zone in these dsDNA substrates bears similarity for the consensus binding behaviors to emerge.

As for recessed dsDNA substrates, the binding affinity with mAPE1 is higher than that of the blunt-ended dsDNA. Most of the matched recessed dsDNA with 20-nt 5′-overhang can form the complex with APE1 at the 5 μM protein concentration, which is not sufficient for APE1 to bind the blunt-ended substrate (Supplementary Fig. 10b). Therefore, the presence of 5′-overhang can affect the binding affinity with APE1 and its nuclease activity as mentioned earlier (Fig. 3b, c). Furthermore, the 2-nt mismatched recessed dsDNA shows a reduced binding affinity with mAPE1 when comparing to the other recessed dsDNAs (Fig. 6d), indicating that the presence of 3′-overhang also affects the interactions with mAPE1.

For the substrates that APE1 binds in the middle as illustrated in earlier structures, AP site-containing dsDNA and 1-nt-gapped dsDNA are synthesized here for measuring the binding affinity with mAPE1 (Fig. 6e, f). On the gel, the band of protein–DNA complex similar to that in the aforementioned substrates is also observed. This band is rather smeared, though, reflecting the likelihood of more than one APE1 attaching to a dsDNA substrate in the case of terminal binding[37]. For the binding of mAPE1 with AP site-containing dsDNA and gapped dsDNA, on the other hand, an additional band for a lower molecular weight

protein–DNA complex is observed at the lower protein concentration of 0.5 μM (Fig. 6e, f). The higher affinity and lower molecular weight indicate binding of a single APE1 in the middle of AP site-containing dsDNA or gapped dsDNA. This result is consistent with the structural analysis of APE1 in complexation with an AP site-containing dsDNA, which shows a higher number of hydrogen bonds between APE1 and the substrate when comparing to the complex structures of end-binding (Fig. 6a). Therefore, mAPE1 has a higher affinity in binding an AP site or a gapped site than that of attaching to dsDNA terminus. For 1-nt mismatched gapped dsDNA, however, the lower molecular weight band does not appear, indicating that base pair mismatching can also affect APE1 binding, at least in gapped dsDNA substrates (Fig. 6f).

Regarding nicked dsDNA with 5′-phosphoryl or 5′-hydroxyl, mAPE1 binding is weaker in the case of 5′-phosphoryl (Supplementary Fig. 10c). This trend is in line with its lower exonuclease activity (Fig. 3e). Having the larger phosphoryl group at the 5′ end may lead to the extra hindrance that affects substrate binding and the exonuclease activity of APE1. For mAPE1 binding nicked dsDNA with 5′-hydroxyl, a weak band corresponding to the lower molecular weight protein–DNA complex as in AP site-containing dsDNA or in matched 1-nt-gapped dsDNA is also observed, but the nicked dsDNA with 5′-phosphoryl does not have this feature. Furthermore, the mAPE1 binding of 5′-hydroxyl nicked dsDNA with 1-nt mismatch does not have this band, either (Supplementary Fig. 10). This result is similar to the case of 1-nt mismatched gapped dsDNA. Together, EMSA measurements of mAPE1 binding with matched dsDNA substrates, including AP site-containing dsDNA, 1-nt-gapped dsDNA, and 5′-hydroxyl nicked dsDNA show the common feature of the lower molecular weight band. However, having a

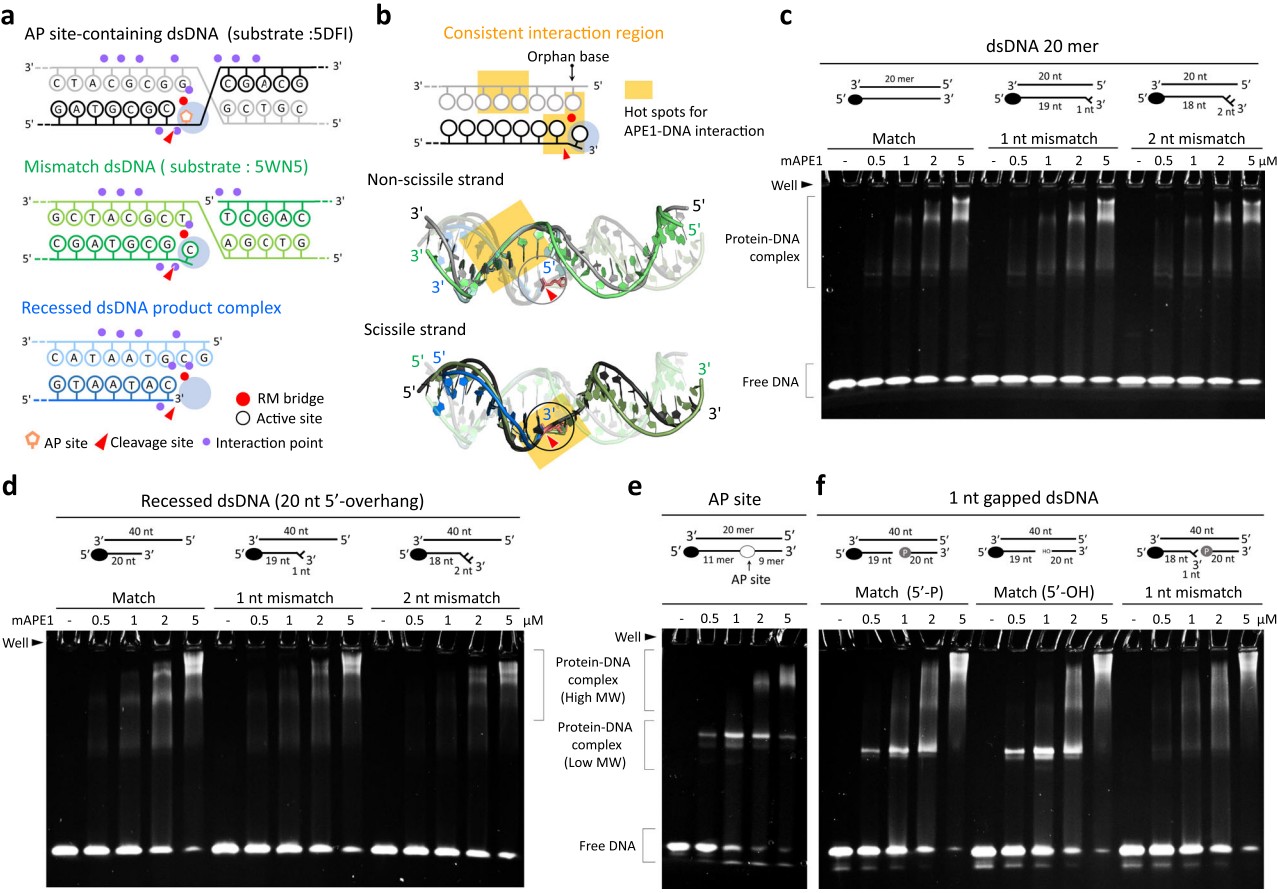

**Fig. 6 Structural analysis and binding assay measurements of APE1 on various dsDNAs. a** The comparison of protein–DNA interacting patterns in hAPE1 AP site-containing dsDNA complex (PDB entry: 5DFI), hAPE1-mismatched dsDNA substrate complex (PDB entry: 5WN5), and mAPE1-recessed dsDNA product complex. The interactions between APE1 and DNA are marked by purple balls. **b** Superposition of the dsDNAs in the aforementioned APE1–dsDNA complexes. The consistent interaction region emerges in the overlay of these structures, yellow boxes. **c–f** EMSA measurements of full-length mAPE1 binding with a match or mismatched blunt-ended dsDNA, recessed dsDNA, AP site-containing dsDNA, and 1-nt gapped dsDNA. The band of protein–DNA complex with a lower molecular weight is specifically observed in the binding with AP site-containing dsDNA and matched 1-nt gapped dsDNA. **c–f** Source data are provided as a Source Data file.

mismatched base pair in these substrates leads to the disappearance of this behavior, indicating altered binding with mAPE1.

## Discussion

By integrating the systematic analysis using nuclease activity assay, binding affinity measurements, X-ray crystallography of APE1 terminal binding, and structural modeling with a molecular mechanical force field, this study uncovers unprecedented knowledge for the exonucleolytic cleavage conducted by APE1. Our results establish that as an exonuclease, APE1 does not favor specific bases but distinguishes dsDNA structures. In particular, APE1 is a poor exonuclease for dsDNAs having a 3′-overhang longer than two nucleotides, but can readily digest matched or 1-nt-mismatched recessed dsDNA, gapped dsDNA, and 5′-hydroxyl nicked dsDNA. In fact, a consistent interaction zone in the dsDNA substrate can be extracted from the combined APE1–DNA complex structures covering both modes of middle binding and end-binding; structures of the latter group only become available until this work. Furthermore, the binding affinity measurements using EMSA corroborate that both the endo- and exonucleolytic cleavage utilize the same interaction sites in APE1. The RM bridge formation induced by the interactions with dsDNA at the active site is discovered here as a key structural

feature for the lack of base specificity in the exonucleolytic digestion of APE1. As an endonuclease, the RM bridge is also found to convey APE1 the AP site specificity. Together, a unified principle of induced space-filling can be deduced from the results of this work to describe the versatile nuclease activities of APE1 and to convey the property of distinguishing substrate structures. In the following, more detailed discussions are provided for these new findings.

Although earlier studies showed that in the middle-binding mode, APE1 can excise the mismatched 3′-end of a dsDNA substrate[20,24,25,37,38], whether the physiologically more relevant matched dsDNA substrates are also vulnerable to the exonuclease activity of APE1 has not been established. Our results show that the matched dsDNA substrates can indeed be digested by mAPE1, albeit they have higher recalcitrance than the mismatched counterpart (Fig. 3). However, when the substrate has a longer 5′-overhang, the activity of APE1 in excising matched dsDNA becomes very close to that in processing the mismatched substrates. Therefore, APE1 can take on the matched as well as mismatched dsDNA encountered in the cell and exert exonuclease activity upon them. The actual catalytic power, though, is sensitive to the structural properties of the substrate.

In the exonucleolytic digestion of dsDNA with matched base pairing, Arg176 interacts with the orphan base and the nearby bases to induce the formation of the RM bridge (Fig. 5b and

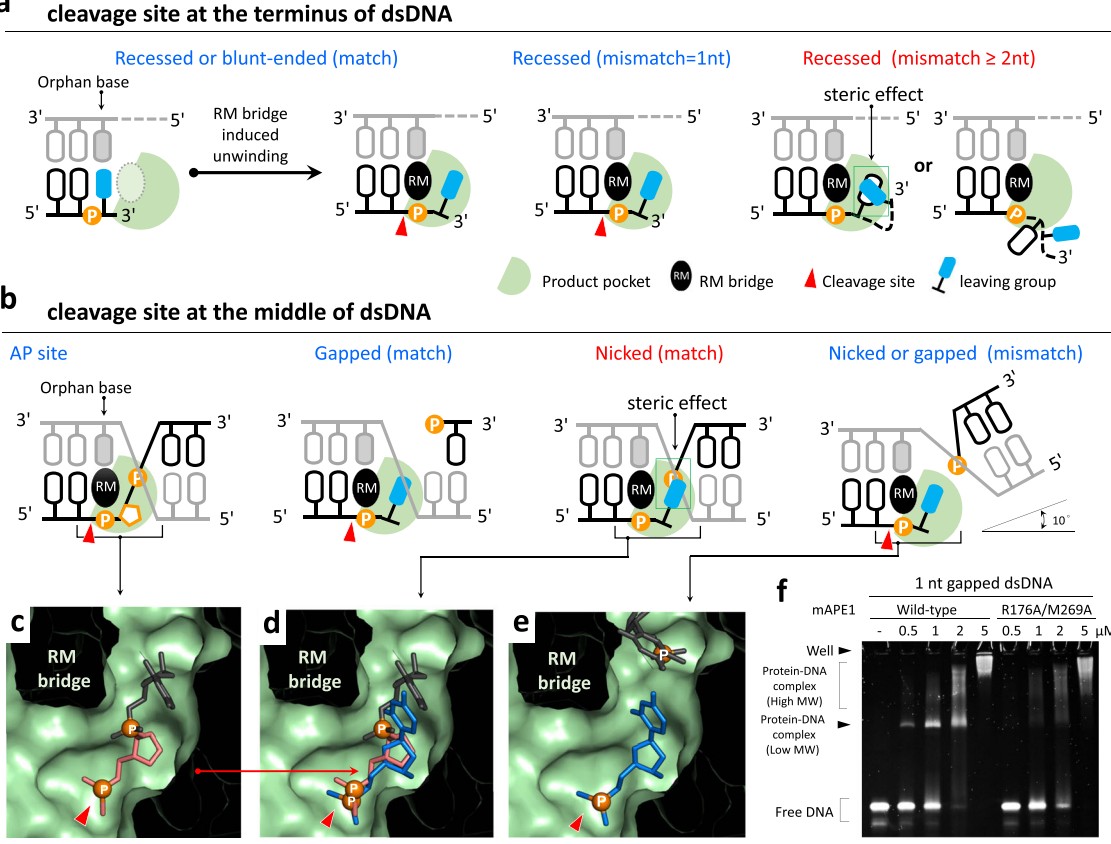

**Fig. 7 The induced space-filling model for APE1 in processing various duplex DNAs.** Based on the positions of cleavage sites in dsDNA substrates, they are classified into two groups, **a** terminal binding and **b** middle binding. **c** The close-up view of the product pocket and AP site-containing dsDNA (PDB entry: 5DFI). Only the AP site (orange), upstream and downstream phosphoryl group, and a downstream nucleotide (black) of the substrate dsDNA is shown. **e** The close-up view of the product pocket and the 1-nt mismatched nicked dsDNA (PDB entry: 5WN5). Only the leaving group (blue) and a downstream nucleotide (black) of the substrate dsDNA is shown. **d** Superposition of the leaving group in 1-nt mismatched nicked dsDNA (blue) and the AP site (orange), upstream and downstream phosphoryl group, and a downstream nucleotide (black) in the AP site-containing dsDNA. The base region of the leaving group is overlapped to the downstream phosphoryl group near the AP site. **f** EMSA measurements of wild-type and double mutant (R176A/M269A) mAPE1 binding with matched 1-nt gapped dsDNA. The band of protein–DNA complex with a lower molecular weight is not observed in the binding of mAPE1 double mutant. **f** Source data are provided as a Source Data file.

Supplementary Fig. 8). The RM bridge can cause the orphan base to flip out the active site and allows the leaving group to enter the product pocket. The RM bridge is thus crucial for APE1 to process matched dsDNA substrates exonucleolytically (Fig. 7a). The cleaved leaving group does not have strong interactions in the product pocket since it readily releases and is absent in the APE1–dsDNA complex structures resolved here.

With the first APE1 structures in terminal complexation with the dsDNA products resolved here, we show that APE1 uses the same cleavage site over the consistent interaction zone in both its endonuclease and exonuclease activities. Superposition of our structures with those of APE1 complexing a substrate or product, including middle-binding with AP site-containing dsDNA (PDB entry: 5DFI and 5DFF)[11] and middle binding with mismatched dsDNA (PDB entry: 5WN5 and 5WN1)[17], indeed shows the same cleavage sites (Supplementary Fig. 11). Combining with the new terminal-binding APE1 structures with matched dsDNA resolved here, this principle is shown to apply to both the cases of matched and mismatched dsDNA substrates. As such, the hAPE1 structure of complexing the 3′-matched dsDNA (PDB entry: 5WN0)[17] in a middle-binding mode is supposed to be the product-bound form after removing the mismatched nucleotide. In summary, APE1 using the same cleavage site over the consistent interaction zone in processing matched and mismatched dsDNA substrates is consistent with all available structural data. Although previous

structural analysis has suggested that the cleavage sites in APE1 for matched and mismatch base pair could be different and might be 7.5 Å apart[17], the mutagenesis experiments conducted here and in previous works[16,39] are in line with the mechanism of consistent interaction zone as discovered here.

APE1 is a versatile nuclease that can process various dsDNAs, including AP site-containing dsDNA, matched, and mismatched dsDNA. After substrate binding, the formation of the RM bridge fills in and divides the active site space into the substrate binding region and the product pocket. In this induced space-filling picture, the RM bridge provides the steric hindrance to make the product pocket long and narrow. For endonucleolytic cleavage, the product pocket can only fit the AP site in the presence of upstream and downstream phosphates (Fig. 7b, c), leading to the high selectivity for this substrate.

For exonucleolytic digestion of blunt-ended dsDNA or recessed dsDNA without the downstream phosphoryl group, the nitrogenous base can fit in the product pocket upon the induced space-filling of the RM bridge (Fig. 7a, d). For a matched dsDNA substrate, the RM bridge can interact with the orphan base and unwinds the base pairing so that the leaving group can move in the product pocket for cleavage. Furthermore, we show that APE1 can bind 2-nt mismatched dsDNA, but cannot cleave the substrate (Figs. 3c and 6d), indicating that the 2-nt long 3′-overhang is too big to fit in the product pocket (Fig. 7a).

For APE1 binding with AP site-containing dsDNA, matched gapped dsDNA, and 5′-hydroxyl-matched nicked dsDNA, a lower molecular weight band is observed in EMSA that does not appear in APE1 binding with mismatched dsDNA substrates. The induced space-filling model of APE1 forming a narrow product pocket also serves to explain this difference between targeting matched and mismatched dsDNA. Using the structure of APE1 complexing an AP site-containing dsDNA in the middle as a template, introducing matched 5′-phosphoryl nicked dsDNA encounters steric hindrance for the downstream phosphoryl group so that it cannot be fit in the product pocket (Fig. 7b, d). This structural feature reflects in the low-binding affinity and activity of APE1 in processing 5′-phosphoryl matched nicked dsDNA. In contrast, 5′-hydroxyl matched nicked dsDNA becomes a feasible substrate since the hydroxyl group is sufficiently small to fit in the product pocket (Fig. 3e and Supplementary Fig. 10c).

For mismatched dsDNA substrates, on the other hand, the structures show that the steric hindrance for the phosphoryl group can be avoided. In the APE1 active site, the downstream region of the mismatched nicked dsDNA bends and shifts ~10°[17] and the steric hindrance between the downstream phosphoryl and the product pocket is avoided (Fig. 7b, e). As such, the 5′-phosphoryl mismatched nicked dsDNA can be processed by APE1 (Fig. 3e). Under the induced space-filling framework, this property of APE1 in distinguishing matched and mismatched base pairing structures is also observed for gapped dsDNA substrates (Figs. 6f and 7b).

The RM bridge composed of Arg176 and Met269 plays multiple roles in the biocatalysis of APE1, including facilitating protein–DNA interactions[10,17,40], controlling product release[10,41], coupling to the orphan base for unwinding the matched base pair as illustrated by the newly solved terminal-binding structures, and providing steric hindrance in the product pocket to distinguish substrates based on the structural comparison analysis conducted here. Comparing to the wild-type APE1, point mutation on Arg176 or Met269 leads to slightly increased endo- and exonuclease activity, and reduced substrate binding ability[10,17,40,41]. Here, we also measure the exonuclease activity of the R176A/M269A double mutant of APE1 in digesting matched dsDNA substrates and the results are similar to those of the single mutants (Supplementary Fig. 12a). However, the APE1 double mutant shows lower binding affinity with the substrates, and the lower molecular weight band of protein–DNA complex is not observed in binding AP site-containing dsDNA and matched 1-nt-gapped dsDNA, unlike the case of wild-type APE1 (Fig. 7f and Supplementary Fig. 12b). These results indicate that the specificity of substrate recognition is diminished without space-filling due to the R176A/M269A double mutation. The slightly increased nuclease activity is likely a result of enhanced product release[10,41] in the absence of the RM bridge.

Interestingly, mutation of Tyr268 (hTyr269) next to Met269 was shown to affect the endonuclease activity of APE1, presumably by alternating the ability to bend DNA and/or modulating protein–DNA interactions, since the flexibility of the Arg176 (hArg177) containing loop appears to be affected by the Tyr268 (hTyr269) mutation in molecular dynamics simulation[35]. This work suggests that the interactions involving Arg176, Tyr 268, and Met269 exhibit complicated patterns in APE1 to impact the endonuclease activity[35]. Furthermore, the role of Met269 (hMet270) in the exonuclease reaction remains ambiguous[17]. Our structural analysis with binding and activity measurements tailored for the exonuclease activity also illustrate the multifaceted involvement of Arg176 and Met269 and the induced space-filling model with RM bridge can serve to explain the data of different substrates for both endo and exonucleolytic cleavage. The spatial

arrangement of these residues in the active site region indeed expects a coupled interaction network for a sequence change to impact functional activities. This complexity highlights the importance of our approach that integrates the novel structural information of different binding modes with molecular modeling and systematic measurements of activity and binding.

Both Arg176 and Met269 are conserved in vertebrate APE1 or Exo III, like human, mouse, and zebrafish[42], but not in bacteria homologs, such as exonuclease III in E. coli[43], ExoA in bacillus subtilis[44], AP endonuclease in Archaeoglobus fulgidus[45], and NApe and Nexo in Neisseria meningitides[46]. Furthermore, the two RM bridge residues do not appear in vertebrate APE2[47]. Conservation of RM bridge residues may reflect in the catalytic properties of APE1- or Exo III-liked nucleases which is critical for deducing their cellular functions. For example, without the RM bridge residues, APE2 bears stronger exonuclease activity and weaker endonuclease activity than APE1 does, and thus APE1 and APE2 can have non-overlapping functions and hence play roles in different steps in SSB[6,47]. Therefore, the difference in catalytic properties due to RM bridge can be utilized for strategic drug design. For example, the presence of RM bridge in APE1 may lead to distinct inhibitors from those of APE2[47], yeast Apn1, and E. coli Nfo[20]. These behaviors highlight the value of our integrated studies with novel structural information of APE1.

## Methods

**Protein expression and purification**. The wild-type and N-terminal truncated mouse APE1 (mAPE1) genes were cloned into a pET28a vector, respectively, and expressed in E. coli BL21-CodonPlus(DE3)-RIPL strain. E. coli cells were cultured in Luria Broth (LB) medium at 37 °C supplemented with 50 mg/ml kanamycin, 35 mg/ml chloramphenicol and 25 mg/ml streptomycin to an $OD_{600}$ of 0.4–0.6 and then induced by 1 mM isopropyl β-D-1-thiogalactopyranoside at 18 °C for 18 h. The cells were collected through centrifugation at 6,721 g for 30 min at 4 °C and further lysed through sonication in 50 mM Tris-HCl pH 8.0, 300 mM NaCl. The cell debris was clarified through centrifugation at 20,216 g at 4 °C for 30 min and the supernatant was loaded into an affinity column (HiTrap TALON crude 5 ml, GE Healthcare) and purified by standard protocol. Target proteins were further purified by an ion-exchange column (HiTrap™ SP HP 5 ml, GE Healthcare) and a size-exclusion column (HiLoad 16/60 Superdex 75 prep grade, GE Healthcare). Purified wild type or truncated mAPE1 was concentrated to at least 10 mg/mL in 50 mM Tris-HCl pH 7.0, 300 mM NaCl, and stored at −20 °C until use. Active site or R176A/M269A mutants were made by QuikChange lightning site-directed mutagenesis kit (Stratagene) and purified by the same procedure.

**Nuclease activity assay**. The sequences of DNA substrates are shown in Supplementary Table 1. DNA substrates were labeled with fluorescein amidite (FAM) at the 5′-end by MDBio, Inc., Taiwan. In endonuclease activity assays, 0.5 μM labeled substrates were mixed with protein in 20 mM Tris-HCl pH 7.0, 120 mM NaCl, 2 mM $MgCl_2$. In exonuclease activity assays, 0.5 μM labeled substrates were mixed with protein in 20 mM Tris-HCl pH 7.0, 30 mM NaCl, 2 mM $MgCl_2$. All reactions were incubated at 37 °C for 30 min and stopped by adding 2xTBE–urea sample loading buffer (G-Biosciences, USA) at 95 °C for 5 min. The DNA digestion patterns were resolved on 20% TBE urea denaturing polyacrylamide gels and visualized by blue light.

**Dynamics simulation and energy minimization of DNA models**. To develop molecular models for the substrate binding of APE1 with a damaged base in the 3′-end, the X-ray structure of mAPE1 blunt-ended dsDNA product complex is used as the template. The CHARMM software and the CHARMM36 protein and nucleic acid force fields[36] are used for all calculations conducted in this work. First, the coordinates of hydrogen atoms in APE1 and DNA are generated using the internal coordinate facility of CHARMM. The force field of each damaged base is generated by analogy using the CGENFF approach[48,49], and the nucleic acid residue with the damaged based is then added to the 3′-end of product DNA in the mAPE1 blunt-ended dsDNA product complex structure. Geometry optimization is then conducted with all heavy atoms of the protein and nucleic acid substrate fixed at the X-ray resolved coordinates. The cutoff radius for non-bound interactions is 14 Å. The objective is to understand the fitness of the damaged substrate to the active site environment.

**Electrophoretic mobility shift assay**. mAPE1 was firstly incubated with 50 mM EDTA, 20 mM Tris-HCl pH 7.0, 120 mM NaCl on ice for 10 min, then mixed with FAM-labeled substrates and incubated on ice for 10 min. The reaction mixture was

added with 6×DNA loading dye (0.35% w/v Orange G, 20% ficoll) and loaded into a TBE native gel. The DNA-binding patterns were visualized by ultraviolet (UV) or blue light.

**Crystallization and crystal structure determination**. Purified wild-type APE1 or a truncated mutant with a deletion of 30 amino acids on the N-terminus (mAPE1Δ30) was mixed with ssDNA (5′-CGTAATACG-3′ or 5′-GCGTAATAC-3′) at a molar ratio of 1:1.2 on ice for 10 min. The protein–DNA mixture was added an equal volume of reservoir solution and then cultured by hanging-drop crystallization at 20 °C. The conditions of crystal culture are listed in Supplementary Table 2. All crystals were cryoprotected by Paraton-N (Hampton Research, USA) for data collection at BL13-C1, BL13-B1, and TPS-05A in NSRRC, Taiwan. All diffraction data were processed by HKL2000, and diffraction statistics are listed in Supplementary Table 3. Structures were solved by the molecular replacement method, and the crystal structure of apo-human APE1 (PDB: 1HD7) was used as the search model by MOLREP of CCP4[50]. The models were built by Coot-0.8.1[51] and refined by PHEXIX-1.9-1692[52]. Diffraction structure factors and structural coordinates were deposited in the RCSB Protein Data Bank with the PDB ID code of 7CD5 for the mAPE1 blunt-ended dsDNA product complex and 7CD6 for mAPE1-recessed-dsDNA product complex.

**Reporting summary**. Further information on research design is available in the Nature Research Reporting Summary linked to this article.

## Data availability

The datasets generated and/or analyzed in this study are available from the corresponding author upon reasonable request. The coordinates of all the structures addressed in this work are available in the Protein Data Bank (wwPDB), including the structures determined in this study (PDB IDs "7CD5" and "7CD6") and in previous works (PDB IDs 1HD7, 5DFI, 5DFF, 5WN0, 5WN1, 5WN4, and 5WN5). Source data are provided with this paper.

## Code availability

Coordinates and structure factors were deposited in Protein Data Bank under accession code 7CD5 (mAPE1 blunt-ended dsDNA product complex) and 7CD6 (mAPE1-recessed-dsDNA product complex).

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

## Acknowledgements

A portion of this research was performed at the National Synchrotron Radiation Research Center (BL-13B1, BL-13C1, and TPS-05A), a national user facility supported by the National Science Council of Taiwan, ROC. The Synchrotron Radiation Protein Crystallography Facility is supported by the National Core Facility Program for Biotechnology. This work is financially supported by the Ministry of Science and Technology (MOST) of Taiwan under grant number MOST 109-2113-M-009-023 to J.-W.C. and through the Excellent Youth Scholar Grant and Young Scholar Fellowship (Columbus) Program under grant number MOST 107-2628-B-009-001, MOST 108-2636-B-009-004, and MOST 109-2636-B-009-004 to Y.-Y.H. The financially support from the "Center For Intelligent Drug Systems and Smart Bio-devices (IDS²B)" and the "Smart Platform of Dynamic Systems Biology for Therapeutic Development" project from The Featured Areas Research Center Program within the framework of the Higher Education Sprout Project by the Ministry of Education (MOE) in Taiwan is also acknowledged. This study is supported partially by the NCTU-KMU JOINT RESEARCH PROJECT, Kaohsiung Medical University (NCTUKMU108-DR-01).

## Author contributions

T.-C.L., C.-T.L., K.-C.C., and K.-W.G. performed experimental studies. J.-W.C. performed computational modeling. T.-C.L., S.W., J.-W.C., and Y.-Y.H. designed experiments and analyzed data. Y.-Y.H. supervised the work.

## Competing interests

The authors declare no competing interests.
