## [Peer Review File · Nature Communications]

REVIEWER COMMENTS

Reviewer #1 (Remarks to the Author):

The manuscript "APE1 Distinguishes DNA Substrates in Exonucleolytic Cleavage by Induced Space-Filling" from Liu et al. presents a structural view of the 3'-exonuclease activity of APE1. While the endonuclease activities of APE1 (removing apurinic/apyrimidinic sites and mismatches from the middle of the DNA duplex) are well characterized, the question of how APE1 binds to the end of the DNA duplex and removes both matched and mismatched bases from its termini remained open. This 3' to 5'-exo activity of APE1 serves as proofreading for polymerase β -catalyzed gap filling during base excision repair.

The authors systematically investigate the 3'-exo and endo- enzymatic activities of mouse APE1 on various DNA substrates. They demonstrate that APE1 digests a 3'-termini of the DNA duplexes with matched, mismatched and damaged DNA bases and prefers substrates with 5'-overhangs.

Further, the authors present two crystal structures of APE1 bound to the end of DNA duplexes. In the first structure a single base from the 3'-end of the blind end DNA duplex got digested during crystallization resulting in the complex with a single base 5'-overhang. The second structure has a two base 5'-overhang. Both structures are of good quality and have reliable electron density maps for the DNA. Both structures superimpose well with human APE1 bound to a gapped DNA duplex (endo-excision product complex) so the 3'-ends of the DNA strands in the 3'-exo complexes and the 3'-end of the gapped DNA strand in the endo complex are positioned similarly relative to the active site of the protein.

Overall, while the data in the manuscript are of sound quality, there is a problem with the interpretation of the structural data. The authors introduce a novel Arg176-Met269 interaction termed "RM bridge" in their two 3'-endo APE1 DNA-complexes and all other determined to date APE1-DNA complexes. They propose that this "RM bridge" is "induced by ds DNA binding" and "may serve to separate the scissile and non-scissile strands to make the orphan base away from the active site and to push the last nucleotide at the 3'-end into the product pocket for cleavage". However, the description of this structural feature was provided two decades ago by J. Tainer group (Nature, 2000, p 451). The loop containing M270 in human APE1 (mouse Met269) intercalates the minor groove of the DNA duplex, while the loop containing human Arg177 (mouse R176) intercalates the duplex from the major groove opposite the hM270 loop. The hM270-hR177 'double loop' was proposed to cap the active site, stabilize the DNA conformation, and secure APE1 onto DNA for catalysis. Despite the apparent structural importance of hM270, its mutation to Ala had little effect on APE1 activity. The recent work from Bret Freudenthal's group (Hoitsma et al, NAR 2020, p7345) shows that hY269 positioned next to hM270 has a critical effect on the APE1 activity. Mouse Ape1 has Y268 in the equivalent position. Thus, the two important questions for the manuscript under review are:

1. Why introduce a new "RM bridge" term for the long-known APE1's structural feature?
2. Given that hY269 (mY268) is more significant for the enzyme's function than hM270 (mM269) is "RM bridge" important?
3. How strong are the evidence for the proposed by the authors "induced space-filling mechanism" for endo- and exo- APE1 catalysis if it is based on the "RM bridge".

Other points:

1. During crystallization of APE1 with the blind-end DNA duplex, one base from the 3'-end got removed forming a DNA duplex product with a single base 5'-overhang. The crystallization conditions listed in Supplementary Table 2 do not include Mg²⁺ ion, which is essential for the catalysis by APE1. There are no other divalent ions in the crystallization conditions to substitute for Mg²⁺ neither. How did the catalysis occur? The authors should provide an explanation.
2. Page 5, line 7 from the bottom. The authors use a term "molecular mechanics modeling". This term is not commonly used and is not defined by the authors.
3. Please, include the B-factors for the protein and DNA into Supplementary Table 3.

Reviewer #2 (Remarks to the Author):

Comments to the Authors:

APE1 is an enzyme that has endonuclease and exonuclease activities. In this manuscript, Liu et al., performed biochemical assays to study the nuclease function of mouse APE1 and obtained two crystal structures of APE1-DNA complexes using blunt-ended-dsDNA and recessed dsDNA. They observed that exonucleolytic function of APE1 does not have sequence specificity, but recognizes substrates through structural features of DNA. The authors also proposed an induced space-filling model, which can be applied to both endonucleolytic and exonucleolytic cleavage. After binding to DNA, APE1 forms a RM bridge that generates a long and narrow product pocket for substrate selection.

Overall, the structural and biochemical data in this work is of high quality. While the data provide some interesting insight into the substrate binding of APE1, there are major comments/concerns that need to be addressed before this manuscript would be suitable for publication:

Major points:

1. The authors determined two structures of APE1-dsDNA complex using matched dsDNA. They claimed that these are the first APE1 structures showing the end-binding. However, it is not clear about the physiological relevance and importance of this end-binding for APE1. Thus, the author should address more about the possibility of APE1 working on the terminal-end of dsDNA.
2. The major functions of APE1 are to do endonucleolytic and exonucleolytic cleavages on AP site and mismatched DNA, respectively. In the previous APE1 study (Nature Comm, 2018), Whitaker et al., already provided detailed molecular mechanisms by determining structures of APE1 bound to a series of dsDNA, including matched and mismatched dsDNA. In these structures, APE1 is bound to the middle of dsDNA, which represents a state closer to the ideal physiological condition. The author should address the physiological importance of terminus-binding and exonucleolytic activity of APE1 on matched DNA so readers would better understand the new findings and importance of this work compared to the previous published study.
3. The PDB (5WN1) is an APE1 exonuclease product complex, in which the mismatched nucleotide was removed. This 5WN1 is similar to the product complex in this work. The 3' nucleotide is removed in the current structure of APE1 bound to blunt-ended dsDNA. The authors should compare two structures and describe structural differences.

Minor points:

1. In Fig 6f, the authors claimed that the condition of 1-nt mismatched nicked dsDNA (the most right five lanes) does not show the Low-MW bands. In Fig. S9C, the authors claimed that the condition of match dsDNA(5'-OH) have low-MW bands. However, in my point of view, the results from both conditions look similar. How to define the appearance of the low-MW band?
2. The author should provide a panel showing the overall structure of APE1, pointing out the redox domain and truncated regions, mutation residues, in this work for readers, who are not familiar with the APE1 structure.
3. The authors described the importance of RM bridge. The omitted (Fo-Fc) map of this RM bridge should be shown in this work.
4. Fig 6b, "Ape1" should be "APE1".
5. In page 11, lines 1 and 3. "overhnag" should be "overhang".
6. In page 31, "The activity of full-length and truncated mAPE1" should be "The activities of full-length and truncated mAPE1"

7. In page 31, "excise" should be "excised"
8. Page 12, line 11 : "3'-termal" should be "3'-terminal"
9. In page 5, line 19: "revolving" should be "resolving"
10. Figure 6a. "5DF1: Substrat" should be "5DF1: Substrate"
11. Figure legend of Fig. S2, "shown" should be "showed"

Point-to-point reply to the reviewers' comments

Manuscript ID: NCOMMS-20-34214-T

Title: APE1 Distinguishes DNA Substrates in Exonucleolytic Cleavage by Induced Space-Filling

Reviewer #1:

The manuscript “APE1 Distinguishes DNA Substrates in Exonucleolytic Cleavage by Induced Space-Filling” from Liu et al. presents a structural view of the 3'-exonuclease activity of APE1. While the endonuclease activities of APE1 (removing apurinic/apyrimidinic sites and mismatches from the middle of the DNA duplex) are well characterized, the question of how APE1 binds to the end of the DNA duplex and removes both matched and mismatched bases from its termini remained open. This 3' to 5'-exo activity of APE1 serves as proofreading for polymerase β -catalyzed gap filling during base excision repair.

The authors systematically investigate the 3'-exo and endo- enzymatic activities of mouse APE1 on various DNA substrates. They demonstrate that APE1 digests a 3'-termini of the DNA duplexes with matched, mismatched and damaged DNA bases and prefers substrates with 5'-overhangs.

Further, the authors present two crystal structures of APE1 bound to the end of DNA duplexes. In the first structure a single base from the 3'-end of the blind end DNA duplex got digested during crystallization resulting in the complex with a single base 5'-overhang. The second structure has a two base 5'-overhang. Both structures are of good quality and have reliable electron density maps for the DNA. Both structures superimpose well with human APE1 bound to a gapped DNA duplex (endo-excision product complex) so the 3'-ends of the DNA strands in the 3'-exo complexes and the 3'-end of the gapped DNA strand in the endo complex are positioned similarly relative to the active site of the protein.

Overall, while the data in the manuscript are of sound quality, there is a problem with the interpretation of the structural data. The authors introduce a novel Arg176-Met269 interaction termed “RM bridge” in their two 3'-endo APE1 DNA-complexes and all other determined to date APE1-DNA complexes. They propose that this “RM bridge” is “induced by dsDNA binding” and “may serve to separate the scissile and non-scissile strands to make the orphan base away from the active site and to push the last nucleotide at the 3'-end into the product pocket for cleavage”.

However, the description of this structural feature was provided two decades ago by J. Tainer group (Nature, 2000, p 451). The loop containing M270 in human APE1 (mouse Met269) intercalates the minor groove of the DNA duplex, while the loop containing human Arg177 (mouse R176) intercalates the duplex from the major groove opposite the hM270 loop. The hM270-hR177 ‘double loop’ was proposed to cap the active site, stabilize the DNA conformation, and secure APE1 onto DNA for catalysis. Despite the apparent structural importance of hM270, its mutation to Ala had little effect on APE1 activity. The recent work from Bret Freudenthal’s group (Hoitsma et al, NAR 2020, p7345) shows that hY269 positioned next to hM270 has a critical effect on the APE1 activity. Mouse Ape1 has Y268 in the equivalent position. Thus, the two important questions for the manuscript under review are:

1. Why introduce a new “RM bridge” term for the long-known APE1’s structural feature?

Our reply:

We thank Reviewer 1 for this question and providing a historic perspective on APE1 structures. As the reviewer pointed out, we present in this work two crystal structures of APE1 bound to the end of DNA duplexes, which are first for the terminal binding mode. On the other hand, in earlier crystal structures as the ones Reviewer 1 mentioned, APE1 is in middle-binding and the discussions therein primarily focused on the endonuclease activity. The APE1 structures we solve here allows putting together both the middle-binding and terminal-binding modes to deduce a comprehensive and unified picture. We find that the dsDNA substrates have very similar structures in the consistent interaction zone but their structures diverge in the product pocket, and the RM interaction appears like a “bridge” separating the two regions. Therefore, the term RM bridge is used to represent this new physical picture revealed with the terminal-binding structures resolved here. The induced space-filling mechanism of the RM bridge also explains the substrate selection data of APE1 in both endo- and exonucleolytic cleavage. In the revised manuscript, extensive edits, addition, and reorganization are made in the Introduction, Results, and Discussion sections to better illustrate these points.

2. Given that hY269 (mY268) is more significant for the enzyme’s function than hM270 (mM269) is “RM bridge” important?

Our reply:

The active site location of hM270 (mM269) and its spatial arrangement of with hArg177 (mArg176) has drawn much attention to analyze its role, but as discussed in Amy M. Whitaker et al., Nat. Commun. 2018, given the previously available structures and mutagenesis data, the role of hM270

(mM269) in the exonuclease reaction remains mysterious. In the recent work of Hoitsma et al, NAR 2020, it was shown that mutation of hTyr 269 (mTyr268) next to hM270 affects the endonuclease activity of APE1, presumably by alternating the ability to bend DNA and/or modulating protein-DNA interactions, since flexibility of the hArg177 containing loop appears to be affected by the hTyr 269 (mTyr268) mutation in molecular dynamics simulation. This earlier work thus suggests that the interactions involving hArg177, hTyr 269, and hMet270 exhibit complicated patterns in APE1 to impact the endonuclease activity.

Consistent with the earlier works, our structural analysis with binding and activity measurements tailored for the exonuclease activity also illustrate multifaceted involvement of mArg176 and mMet269 and the induced space-filling model with RM bridge can serve to explain the data of different substrates for both endo and exonucleolytic cleavage. In particular, the binding measurement of the R176A/M269A mutant shows that the RM bridge regulates the specificity of APE1 substrate selection, highlighting its importance. The spatial arrangement of these residues in the active site region indeed expects a coupled interaction network for a sequence change to impact functional activities, and molecular processes such as enhanced product release (as addressed in the Discussion section at page 21 of the revised manuscript) would also come to play. In the revised manuscript, we extensively reorganize and expand the Discussion section (page 21 and 22) to make clear these points.

3. How strong are the evidence for the proposed by the authors “induced space-filling mechanism” for endo- and exo- APE1 catalysis if it is based on the “RM bridge”.

Our reply:

Comparing apo and dsDNA-bound APE1 structures provides strong evidence that the RM bridge is induced by the interactions with the bound substrate. Our integrated structural analysis including the end-binding mode then shows that the induced-space filling of the RM bridge separates the active site region into consistent interaction zone and the narrow product pocket. The steric hindrance provided by the latter is able to explain the diverse behaviors of substrate selectivity for both the endo- and exonuclease activity. This model is tested against the activity and binding assay data of targeting AP site contained dsDNA for endonucleolytic cleavage and the data of targeting dsDNA with various damaged bases as well as matched/mismatched gapped, nicked, blunt-ended, and recessed dsDNA for exonucleolytic cleavage. These results thus provide very strong evidence for the induced-space filling mechanism, and its ability to explain such diverse behaviors is unprecedented. In the revised manuscript, extensive edits and reorganization are made in the Results and Discussion sections to better illustrate these points.

Other points:

1. During crystallization of APE1 with the blind-end DNA duplex, one base from the 3'-end got removed forming a DNA duplex product with a single base 5'-overhang. The crystallization conditions listed in Supplementary Table 2 do not include Mg²⁺ ion, which is essential for the catalysis by APE1. There are no other divalent ions in the crystallization conditions to substitute for Mg²⁺ neither. How did the catalysis occur? The authors should provide an explanation.

Our reply:

Given that the time of crystal growth can be as long as three to five weeks, a trace amount of endogenous divalent ions from the *E. coli* host would be sufficient to render activity for APE1. Similar behavior has also been observed in many other exonuclease systems (Huang, K. W. et al., PLOS Biol. 2018 and Hsiao Y. Y. et al., PLOS Biol. 2014). In the Results section of the revised manuscript, discussion is added to address this point (page 10).

2. Page 5, line 7 from the bottom. The authors use a term “molecular mechanics modeling”. This term is not commonly used and is not defined by the authors.

Our reply:

In the revised manuscript, this term is changed to “structural modeling based on an empirical force field of molecular mechanics,..”

3. Please, include the B-factors for the protein and DNA into Supplementary Table 3.

Our reply:

Following the reviewer’s suggestion, the B-factors of protein and DNA are included in Supplementary Table 3 of the revised manuscript.

Reviewer #2

Comments to the Authors:

APE1 is an enzyme that has endonuclease and exonuclease activities. In this manuscript, Liu et al., performed biochemical assays to study the nuclease function of mouse APE1 and obtained two crystal structures of APE1-DNA complexes using blunt-ended-dsDNA and recessed dsDNA. They observed that exonucleolytic function of APE1 does not have sequence specificity, but recognizes substrates through structural features of DNA. The authors also proposed an induced space-filling model, which can be applied to both endonucleolytic and exonucleolytic cleavage. After binding to DNA, APE1

forms a RM bridge that generates a long and narrow product pocket for substrate selection.

Overall, the structural and biochemical data in this work is of high quality. While the data provide some interesting insight into the substrate binding of APE1, there are major comments/concerns that need to be addressed before this manuscript would be suitable for publication:

Major points:

1. The authors determined two structures of APE1-dsDNA complex using matched dsDNA. They claimed that these are the first APE1 structures showing the end-binding. However, it is not clear about the physiological relevance and importance of this end-binding for APE1. Thus, the author should address more about the possibility of APE1 working on the terminal-end of dsDNA.

Our reply:

In the end-binding mode, the 3'-end of dsDNA substrate is exposed for exonucleolytic digestion by APE1. Such 3'-end exposed dsDNA intermediates are observed in many different DNA repair pathways, including base excision repair (BER) pathway, DNA mismatch repair, nucleotide incision repair (NIR), trinucleotide repeat (TNR) expansion related BER, DNA single-strand breaks (SSB), removal of 3'-blocking groups in a nucleotide excision repair (NER) independent pathway, and apoptosis (as discussed in page 3 and 4 of the revised manuscript). Our design and synthesis of various dsDNA substrates indeed are to mimic these structures. In the Introduction and Results sections of the manuscript, these points are further emphasized.

2. The major functions of APE1 are to do endonucleolytic and exonucleolytic cleavages on AP site and mismatched DNA, respectively. In the previous APE1 study (Nature Comm, 2018), Whitaker et al., already provided detailed molecular mechanisms by determining structures of APE1 bound to a series of dsDNA, including matched and mismatched dsDNA. In these structures, APE1 is bound to the middle of dsDNA, which represents a state closer to the ideal physiological condition. The author should address the physiological importance of terminus-binding and exonucleolytic activity of APE1 on matched DNA so readers would better understand the new findings and importance of this work compared to the previous published study.

Our reply:

As mentioned in the response of the previous question, numerous physiologically important dsDNA intermediates require APE1 to conduct terminal binding, and among them, matched 3'-end is very

commonly encountered. Structural basis in this regard, though, is lacking, and the previous attempts in developing molecular level understanding for the exonucleolytic activity of APE1 on matched dsDNA substrates are thus severely limited. In the Introduction section of the revised manuscript, these points are further elaborated to better reveal the novelty and importance of the current work. Furthermore, in the Results section of the revised manuscript, a statement is added to highlight that the structural dsDNAs designed and experimented in this work are similar to the DNA intermediates in various DNA repair pathways or apoptosis. For example, gapped dsDNA would appear in the TNR expansion related BER, SSB, and NIR; blunt-ended or recessed dsDNA would come out of DNA fragmentation during apoptosis (page 8).

3. The PDB (5WN1) is an APE1 exonuclease product complex, in which the mismatched nucleotide was removed. This 5WN1 is similar to the product complex in this work. The 3' nucleotide is removed in the current structure of APE1 bound to blunt-ended dsDNA. The authors should compare two structures and describe structural differences.

Our reply:

As the reviewer pointed out, the 5WN1 structure is in middle-binding with mismatched dsDNA, similar to the other available DNA complexed APE1 structures in PDB. The newly resolved structures here, on the other hand, are terminal-binding with matched dsDNA substrates. These structures show the very similar substrate positions in the consistent interaction zone but differ in the dsDNA configurations in the product pocket (as discussed in page 14, and Fig 6a and 6b of the revised manuscript). Together, they lead to the induced-space filling model with RM bridge that can explain the specificity toward substrate structures for both endo- and exonuclease activities. In the revised manuscript, extensive edits, addition, and reorganization are made in the Introduction, Results, and Discussion sections to better illustrate these points.

Minor points:

1. In Fig 6f, the authors claimed that the condition of 1-nt mismatched nicked dsDNA (the most right five lanes) does not show the Low-MW bands. In Fig. S9C, the authors claimed that the condition of match dsDNA(5'-OH) have low-MW bands. However, in my point of view, the results from both conditions look similar. How to define the appearance of the low-MW band?

Our reply:

By comparing the bands obtained in APE1 binding for a particular structural dsDNA, the low-MW

band contrasts to the high-MW band to have the appearance of being more concentrated, having less smear, and showing up at a lower protein concentration. For matched nicked dsDNA with 5'-hydroxyl group, the low-MW band in Fig. S9C (Fig. 10C in the revised manuscript) is clearer than that of the other types of nicked dsDNA, and is thus considered “a weak band corresponding to the lower molecular weight protein-DNA complex...” as stated in the Results section main text (page 16). For the case of gapped dsDNA, the mismatched low-MW band is considerably weaker than the matched one (Fig. 6f), indicating the different binding modes with APE1 for the two types of gapped dsDNA.

2. The author should provide a panel showing the overall structure of APE1, pointing out the redox domain and truncated regions, mutation residues, in this work for readers, who are not familiar with the APE1 structure.

Our reply:

Following this excellent suggestion of the reviewer, Figure 4 in the revised manuscript is modified to include the overall structure of APE1 with the information of different components highlighted.

3. The authors described the importance of RM bridge. The omitted (Fo-Fc) map of this RM bridge should be shown in this work.

Our reply:

Following the reviewer's suggestion, the omit map of Arg176 and Met269 is provided in Supplementary Fig. 7 in the revised manuscript. Furthermore, a statement is added in the Results section to refer to the omit map of the RM bridge (page 12).

4. Fig 6b, “Ape1” should be “APE1”.

5. In page 11, lines 1 and 3. “overhnag” should be “overhang”.

6. In page 31, “The activity of full-length and truncated mAPE1” should be “The activities of full-length and truncated mAPE1”

7. In page 31, “excise” should be “excised”

8. Page 12, line 11 : “3'-termal” should be “3'-terminal”

9. In page 5, line 19: “revolving” should be “resolving”

10. Figure 6a. “5DF1: Substrat” should be “5DF1: Substrate”

11. Figure legend of Fig. S2, “shown” should be “showed”

Our reply:

The other grammatical issues raised by the reviewer have been corrected in the revised manuscript.

REVIEWERS' COMMENTS

Reviewer #1 (Remarks to the Author):

The revised version of the manuscript "APE1 Distinguishes DNA Substrates in Exonucleolytic Cleavage by Induced Space-Filling" is significantly improved and adequately addresses the reviewers' critique. The proposed function of the RM bridge in the catalytic mechanism of APE1 is now more clearly described and justified.

Minor comments:

Page 3, last paragraph. "APE1 can directly interact with the error prone polymerase, DNA polymerase β ..". The authors probably wanted to state: "APE1 can directly interact with the error prone β ..".

Page.4, lane 5. "unprecedented structural information". The wording should be toned down.

Reviewer #2 (Remarks to the Author):

In this revised manuscript, the authors have addressed the previous comments. The previous concerns have been adequately addressed. This work is well done. I have no further comments.

Point-to-point reply to the reviewers' comments

Manuscript ID: NCOMMS-20-34214-A

Title: APE1 Distinguishes DNA Substrates in Exonucleolytic Cleavage by Induced Space-Filling

Reviewer #1:

The revised version of the manuscript “APE1 Distinguishes DNA Substrates in Exonucleolytic Cleavage by Induced Space-Filling” is significantly improved and adequately addresses the reviewers' critique. The proposed function of the RM bridge in the catalytic mechanism of APE1 is now more clearly described and justified.

Minor comments: Page 3, last paragraph. “APE1 can directly interact with the error prone polymerase, DNA polymerase β ..”. The authors probably wanted to state:” APE1 can directly interact with the error prone β ...

Our reply: In the last paragraph of Page 3 in the revised manuscript, the statement has been changed according to the reviewer's suggestion.

Minor comments:Page.4, lane 5. “unprecedented structural information”. The wording should be toned down.

Our reply: In line 5 of page 4 in the revised manuscript, the aforementioned statement has been toned down according to the reviewer's suggestion.

Reviewer #2:

In this revised manuscript, the authors have addressed the previous comments. The previous concerns have been adequately addressed. This work is well done. I have no further comments.

Our reply: We thank the reviewer for the very encouraging comment.